# A Cross Modal Knowledge Distillation & Data Augmentation Recipe for Improving Transcriptomics Representations through Morphological Features

**Ihab Bendidi** [1 2 3]  **Yassir El Mesbahi** [1 2]  **Alisandra K. Denton** [1 2]  **Karush Suri** [1 2]  **Kian Kenyon-Dean** [2]
**Auguste Genovesio** [3 *]  **Emmanuel Noutahi** [1 2 *]

[1]Valence Labs, Montréal, Canada
[2]Recursion, Salt Lake City, USA
[3]Ecole Normale Supérieure PSL, Paris, France

Correspondence to: <auguste.genovesio@ens.psl.eu>, <emmanuel@valencelabs.com>.

## Abstract

Understanding cellular responses to stimuli is crucial for biological discovery and drug development. Transcriptomics provides interpretable, gene-level insights, while microscopy imaging offers rich predictive features but is harder to interpret. Weakly paired datasets, where samples from different modalities are not from the same biological replicate but share key metadata such as cell line and perturbation, enable multimodal learning but are scarce, limiting their utility for training and multimodal inference. We propose a framework to enhance transcriptomics by distilling knowledge from microscopy images. Using weakly paired data, our method aligns and binds modalities, enriching gene expression representations with morphological information. To address data scarcity, we introduce (1) *Semi-Clipped*, an adaptation of CLIP for cross-modal distillation using pretrained foundation models, achieving state-of-the-art results, and (2) *PEA* (**P**erturbation **E**mbedding **A**ugmentation), a novel augmentation technique that enhances transcriptomics data while preserving inherent biological information. These strategies improve the predictive power and retain the interpretability of transcriptomics, enabling rich unimodal representations for complex biological tasks.

*Proceedings of the $42^{nd}$ International Conference on Machine Learning*, Vancouver, Canada. PMLR 267, 2025. Copyright 2025 by the author(s).

## 1. Introduction

Understanding how cells respond to various stimuli is fundamental to uncovering cellular functions and identifying novel drug targets. However, current technologies are limited in capturing the full range of cellular activities under diverse conditions, especially given the immense complexity of biological systems (Conesa et al., 2016; Kharchenko, 2021). For instance, the interaction of over 20,000 human protein-coding genes with the estimated $10^{60}$ possible chemical compounds (Reymond, 2015) far exceeds manual analysis, necessitating computational methods. Advances in deep learning for biology, such as predicting protein structures (Jumper et al., 2021), modeling molecular binding (Corso et al., 2023; Evans et al., 2021), and uncovering biological patterns through microscopy and gene expression data (Kraus et al., 2024; Bendidi et al., 2024b; Bourriez et al., 2024), offer powerful tools to address this challenge from a unimodal perspective. However, separately modelling data from various *omics* modalities, such as morphological features, proteomics, and transcriptomics provides unique yet partial insights into cellular behavior (Miao et al., 2021; Carpenter et al., 2006; Lopez et al., 2018). By combining these perspectives through multimodal fusion, researchers can construct more comprehensive representations of biological systems (Lu et al., 2021; Rosen et al., 2023), revealing connections critical for accelerating drug discovery. However, collecting multimodal data paired at the sample level remains infeasible currently due to massive experimental costs and technical challenges.

Given the challenges of collecting fully paired data across biological modalities, our focus is on weakly paired datasets, where clusters of samples from two modalities share a common biological state. In this setting, two samples from different modalities are considered "paired" if they belong

to the same biological state or metadata (Xi et al., 2024). In our case, this means transcriptomics and microscopy imaging samples that are not from the same biological replicate but share the same cell line and were exposed to the same perturbation. However, even weakly paired datasets remain scarce due to the cost and complexity of aligning states across modalities. Only a few such datasets exist, limiting their utility for training or fine-tuning models and making simultaneous inference on both modalities, for multimodal fusion for example, impossible to scale with current resources. To address these constraints, we aim to train models using the limited weakly paired data from transcriptomics and microscopy imaging, while enabling them to operate on a single modality, transcriptomics, during inference. This approach leverages the complementary strengths of the modalities: microscopy images are rich in visual phenotypic features with strong predictive power but are challenging to interpret, while transcriptomics data suffers from weaker predictive power, but is more directly interpretable at the gene level, making it easier to connect to biological mechanisms (Kraus et al., 2024; Bendidi et al., 2024b). The complementarity between these modalities motivates the development of strategies to transfer the rich phenotypic insights from microscopy into transcriptomics representations.

To overcome pairing scarcity in training, we propose two practical solutions: cross-modal knowledge distillation and biologically inspired data augmentation. Knowledge distillation facilitates the transfer of information from one modality to another to enhance its utility. For instance, the predictive strength of morphological features in microscopy images can enrich transcriptomics representations, making them more powerful for downstream tasks like drug discovery (Kraus et al., 2024; Replogle et al., 2022; Ye et al., 2018; Chandrasekaran et al., 2023; Bourriez et al., 2024; Sanchez et al., 2025). However, most distillation techniques rely on supervised objectives, which require precise labels that are often unavailable for most biological modalities. Alternatively, unsupervised alignment methods aim to uncover shared structures between modalities, though this is challenging due to the distinct biological relationships each captures (Appendix Figure 6). We instead propose to leverage alignment techniques for cross-modal distillation by binding transcriptomics to frozen morphological representations. We further introduce a novel biologically inspired data augmentation technique tailored for transcriptomics vectors, which preserves biological information while introducing meaningful variation to the dataset. This augmentation approach addresses the scarcity of paired data by improving the richness and robustness of transcriptomics representations, enhancing their predictive power while retaining their inherent interpretability. By combining these strategies, our framework enriches gene expression representations, offering deeper insights into biological processes and expanding their utility across diverse applications.

To summarize, we introduce in this work a recipe for transferring knowledge from morphological features to transcriptomics representations in weakly paired datasets, composed of the following contributions :

- We present *Semi-Clipped*, a straightforward adaptation of CLIP (Radford et al., 2021) that leverages pretrained large unimodal foundation models with trainable adapters. It achieves state-of-the-art performance in cross-modal distillation under data-scarce conditions for our biological modalities.

- We introduce *PEA*, **P**erturbation **E**mbedding **A**ugmentation, a novel biologically inspired data augmentation technique for representations of transcriptomics, that introduce significant variation in the training data while retaining meaningful biological information of each sample. PEA improves cross-modal distillation in our low data regime and widely outperforms existing augmentation techniques at uncovering novel biological relationships.

## 2. Related Works

**Cross-Modal Knowledge Distillation.** Knowledge distillation transfers knowledge from a teacher model to a student by aligning output distributions, typically using Kullback–Leibler (KL) divergence (Hinton et al., 2015). Variants introduce gradient similarity (Zhu & Wang, 2021), correlation (Huang et al., 2022), or structural losses (Park et al., 2019). Cross-modal methods leverage strong modalities to guide weaker ones, often relying on label information (Gupta et al., 2016; Roheda et al., 2018; Xue et al., 2021; Lee et al., 2023). For instance, C2KD (Huo et al., 2024) uses an online filtering mechanism for soft label alignment, while SHAKE (Li & Zhe, 2022) employs shadow adapters for bidirectional distillation. XKD (Sarkar & Etemad) combines self-supervised learning with cross-modal distillation but requires large paired datasets. To our knowledge, no distillation approach has effectively leveraged unsupervised cross-modal alignment in the context of limited weakly paired data.

**Multimodal Learning.** Multimodal learning encompasses approaches for aligning or merging data types for robust inference. CLIP (Radford et al., 2021) aligns image and text into a shared space, while CSA (han Li et al., 2024) uses pretrained unimodal models for few-shot alignment. Methods like SigClip (Zhai et al., 2023), VICReg (Bardes et al., 2022), and DCCA (Lan et al., 2020) enhance alignment through self-supervised learning or correlation maximization but depend on significant shared information (Tsai

et al., 2021). Multimodal distillation approaches (Yang et al., 2024; Wu et al., 2023; Fang et al., 2021; Wang et al., 2022) typically require paired data and focus on multimodal-to-multimodal distillation and do not leverage multimodal alignment for cross-modal distillation when only one modality is available at inference. Relevant to our setting, (Hager et al., 2023) proposed a contrastive learning framework combining images and tabular data, structurally similar to our microscopy-transcriptomics setting, demonstrating the viability of such combinations for predictive and interpretable medical tasks.

**Biologically Relevant Representations.** Advances in microscopy imaging models have driven progress in high-content screening (Kraus et al., 2024; Yao et al., 2024; Kenyon-Dean et al., 2025; Wenkel et al., 2025), histopathology (Saillard et al., 2024; Chen et al., 2024; Vorontsov et al., 2024), and specialized architectures (Bourriez et al., 2024; Pham & Plummer, 2024). In transcriptomics, foundation models (Cui et al., 2024; Yang et al., 2022; Theodoris et al., 2023; Wen et al., 2024) show promise but often underperform simpler models in biologically relevant tasks, with scVI (Lopez et al., 2018) as an exception (Liu et al., 2023; Bendidi et al., 2024b). Microscopy imaging complements transcriptomics (Camunas-Soler, 2024), but while unimodal datasets (Replogle et al., 2022; Chandrasekaran et al., 2023; Fay et al., 2023) are growing, weakly paired multimodal datasets remain scarce. Recent methods (Xi et al., 2024; Watkinson et al., 2024; Sanchez-Fernandez et al., 2023; Xie et al., 2023) address this kind of limitation for different modalities by leveraging weak pairings through pretrained models with trainable adapters (Fradkin et al., 2024).

**Data Augmentations for Biology.** Data augmentations are crucial in addressing data scarcity for biology, as biologically meaningful augmentations can stabilize and improve performance with limited biological datasets (Moutakanni et al., 2024; Bendidi et al., 2023; 2024a). In computational biology, image augmentations have typically focused on basic transformations like rotations (Alfasly et al., 2024; Lafarge & Koelzer, 2022) or differentiable techniques using adversarial learning for domain generalization (Ruppli et al., 2022; Zhou et al., 2024). For transcriptomics, data being in a representation format allows leveraging existing representation-level augmentation methods (DeVries & Taylor, 2017; Li et al., 2022), though their efficacy for biological contexts remains uncertain. Recently, new biologically inspired techniques have emerged specifically for augmenting transcriptomics and biological representations (Kircher et al., 2022; Li et al., 2023; Nouri, 2025).

## 3. Proposed Approach

**Problem Formulation.** We consider two biological data modalities: a teacher modality $T$ and a student modality $S$, each offering distinct perspectives on cellular behavior. Let $\mathcal{X}_T$ and $\mathcal{X}_S$ represent the datasets from these modalities. The samples $x_T^{(i)} \in \mathcal{X}_T$ and $x_S^{(i)} \in \mathcal{X}_S$ correspond to the same biological perturbation and cell type but are not strongly paired due to biological variability. Each sample is annotated with weak labels $p$ (perturbation) and $l$ (cell type). Both datasets are organized into biological batches $\mathcal{B}_T = \{b_{T,1}, b_{T,2}, \ldots, b_{T,|\mathcal{B}_T|}\}$ and $\mathcal{B}_S = \{b_{S,1}, b_{S,2}, \ldots, b_{S,|\mathcal{B}_S|}\}$. Each batch $b_{T,k} \in \mathcal{B}_T$ and $b_{S,m} \in \mathcal{B}_S$ consists of a set of samples $\{x_{T,k}^{(j)}\}_{j=1}^{N_{T,k}}$ and $\{x_{S,m}^{(j)}\}_{j=1}^{N_{S,m}}$, and each batch includes, in addition to perturbed samples, a number of control (unperturbed) samples, denoted by $\{x_{T,k}^{(c)}\}_{c=1}^{C_{T,k}}$ and $\{x_{S,m}^{(c)}\}_{c=1}^{C_{S,m}}$ for $C_{M,k} \geq 2$.

**Proposed Distillation Method.** Given the scarcity of weakly paired data, we adopt pretrained and frozen unimodal encoders $E_T : \mathcal{X}_T \to \mathbb{R}^{d_T}$ and $E_S : \mathcal{X}_S \to \mathbb{R}^{d_S}$, following (Fradkin et al., 2024). These encoders produce embeddings $z_T^{(i)} = E_T(x_T^{(i)})$ and $z_S^{(i)} = E_S(x_S^{(i)})$ for the teacher and student modalities, respectively. Our objective is to learn a mapping function $f_S : \mathbb{R}^{d_S} \to \mathbb{R}^{d_T}$ that aligns student embeddings to the teacher embedding space, yielding transformed embeddings $h_S^{(i)} = f_S(z_S^{(i)})$ that integrate properties from the teacher modality $T$. We aim to achieve the dual objective of leveraging weak biological labels for pairing while minimizing reliance on them as learning objectives, since such labels underperform compared to unsupervised objectives in microscopy imaging (Kraus et al., 2024), and preventing mutual drift between modalities with limited shared information. We propose *Semi-Clipped*, a straightforward adaptation of the CLIP loss (Radford et al., 2021) for cross-modal knowledge distillation. This approach uses the frozen unimodal encoders to generate embeddings $z_T$ and $z_S$. The teacher representation $z_T$ is fixed, while an adapter function $f_S$ is trained on the student modality by optimizing the CLIP loss between $h_S$ and $z_T$ to produce aligned embeddings $h_S$. By freezing the teacher embedding space, this avoids dependence on massive amounts of paired data for encoder training and ensures one-way knowledge transfer from the teacher to the student, mitigating mutual drift and feedback from the student to the teacher.

**Batch Correction for Data Augmentation.** In biological datasets, batch effects, or variability caused by differences in experimental conditions, introduce noise that can obscure meaningful patterns. Traditional batch correction techniques (Bendidi et al., 2024b; Celik et al., 2024; Ando et al., 2017) address this by centering embeddings on control (unperturbed) samples within each batch, reducing

noise while preserving the signal. Typically used as a post-processing step, these corrections shift the embedding distribution while retaining key information for downstream analysis. To tackle the scarcity of paired biological modalities, we introduce *PEA* (Perturbation Embeddings Augmentation), a novel biologically inspired augmentation technique that repurposes batch correction as a data augmentation applied directly to the student embeddings during training. Specifically, a function $A : (\mathbb{R}^{d_S}, X_S^c) \to \mathbb{R}^{d_S}$ is randomly selected from a set $\mathcal{A}$ of batch correction transformations and applied to the student embeddings $z_S^{(i)}$. Augmented embeddings $z_{S,A}^{(i)} = A(z_S^{(i)}, X_S^{(c)})$ are then passed to the student adapter $f_S$ for cross-modal knowledge distillation. To ensure the teacher embeddings focus on relevant information, a fixed batch correction $B$ (Ando et al., 2017) is applied to the teacher modality.

To augment transcriptomics data while preserving biological relevance, we extend traditional batch correction techniques into a stochastic augmentation framework. Specifically, for each sample, we randomly select one batch correction transformation $A$ from a predefined set of normalization techniques, ensuring controlled variability in perturbation embeddings. Each selected transformation $A : \mathbb{R}^{d_S} \to \mathbb{R}^{d_S}$ falls into one of these categories: (1) centering, which shifts embeddings by subtracting batch-wise control means to remove batch-specific offsets; (2) scaling, which normalizes variance across features to enhance comparability; and (3) principal component-based transformations that reweight variance along principal axes, emphasizing biologically relevant information while reducing batch artifacts. To introduce further stochasticity, we drop a random subset of the steps of each correction method per sample rather than always applying them sequentially. Additionally, the number of control samples used for correction is randomly sampled per training sample, increasing diversity and robustness to out-of-domain experimental shifts in the learned distributions. Detailed implementation of our batch correction techniques is provided in Appendix Section C.

This method introduces controlled and diverse distributional shifts, helping $f_S$ learn robust, biologically meaningful representations by ignoring batch-induced variability. During inference, a batch correction is applied to the student embeddings $z_S$ to align them with the training distribution, further improving robustness. Algorithm 1 details the process, ensuring the adapter captures biologically relevant features while increasing training diversity and preserving biological information in low-data settings.

## 4. Experimental Setup

**Data & Model Training.** We use microscopy imaging as the teacher modality and transcriptomics as the student modality. The training dataset includes 130,000 ar-

---

**Algorithm 1** Semi-Clipped with PEA implementation

---

**for** each batch $(x_S, x_T) \in (X_S, X_T)$ **do**

  **Extract** using frozen encoders $z_S = E_S(x_S)$ and $z_T = E_T(x_T)$

  **Sample** batch correction function $A \sim \mathcal{A}$

  **Drop** a random subset of steps in $A \to A'$

  **Sample** a random subset of control samples $X_S^{(c)}$

  **Apply** batch correction: $z_S^a = A'(z_S, X_S^{(c)})$

  **Compute** transformed embeddings: $h_S = f_S(z_S^a)$

  **Apply** TVN correction to teacher embeddings: $z_S^b = B(z_T)$ and compute CLIP loss :

$$\mathcal{L} = -\sum_{i=1}^{B} \log \frac{\exp(\text{sim}(h_S^{(i)}, z_T^{(b,i)})/\tau)}{\sum_{j=1}^{B} \exp(\text{sim}(h_S^{(i)}, z_T^{(b,j)})/\tau)}$$

  **Backpropagate** loss $\mathcal{L}$ and update adapter $f_S$
**end for**

---

rayed bulk transcriptomics samples (*HUVEC-CMPD*) and 20,000 microscopy images of human umbilical vein endothelial cells (HUVEC), cells from cell painting, both covering 1,700 chemical perturbations at three concentrations. Each transcriptomics sample can pair with multiple imaging samples based on treatment and concentration, with one pair randomly selected per epoch. These pairs are weakly paired—i.e., they do not originate from the same biological replicate but share the same cell line and perturbation metadata, ensuring comparable biological states across modalities. For encoding microscopy images, we use the pretrained Phenom-1 model (Kraus et al., 2024), a state-of-the-art pretrained model trained on 93 million microscopy images. For transcriptomics, we compare three models: a simple scVI-like MLP[1] trained from scratch on the *HUVEC-CMPD* bulk dataset, scVI (Lopez et al., 2018), a model known for strong performance on small datasets, outperforming existing transcriptomics pretrained models (Bendidi et al., 2024b), and similarly trained from scratch on the *HUVEC-CMPD* bulk dataset, and a pretrained scGPT (Cui et al., 2024) (a pretrained model trained on 33 million transcriptomics samples). A three-layer MLP adapter $f_S$ (input size $d_S$, output size $d_T$) with ReLU activations is trained for the student modality, while both encoders remain frozen. Consistent with (Kenyon-Dean et al., 2025), control samples are excluded from paired data for knowledge distillation and only used for batch correction. The adapter is trained with a temperature of 0.1, learning rate of 0.001, batch size of 1,024, and over 150 epochs.

---

[1]scVI is an MLP-based VAE conditionned on a batch label.

**Evaluation Setting.** The evaluation focuses on assessing the quality of transcriptomic representations after knowledge distillation, emphasizing biological relevance and interpretability. We use a hierarchical benchmarking framework for transcriptomic representations (Bendidi et al., 2024b) (Appendix Section B) with two primary tasks: (1) *Retrieval of known biological relationships*, this task evaluates the ability of the learned representations to capture established biological relationships by retrieving known interactions between genes. Using cosine similarity of gene embeddings, predicted relationships are validated against annotations from CORUM, HuMAP, StringDB, Reactome, and SIGNOR databases. Success is measured by recall scores averaged across these databases, reflecting how well the representations align with known biology. (2) *Transcriptomic interpretability preservation*, this task measures how well the distilled embeddings retain information necessary for reconstructing original gene expression profiles. It evaluates two complementary metrics: the Structural Integrity score, which quantifies how accurately the model preserves the relationships between control and perturbation samples, and the Spearman correlation, which assesses the rank-based agreement between predicted and true gene expression profiles. The average of these metrics provides a comprehensive measure of interpretability preservation.

Success is defined as improving retrieval scores while maintaining interpretability metrics comparable to unimodal transcriptomic representations. This dual focus ensures that the student representations do not collapse or lose transcriptomic-specific information by ignoring it and relying solely on morphological features. Using these tasks, we compare our *Semi-Clipped* approach, with and without PEA, against standard multimodal alignment and cross-modal knowledge distillation methods. For alignment, we include *CLIP* (Radford et al., 2021), *SigClip* (Zhai et al., 2023), *VICReg* (Bardes et al., 2022), and *DCCA* (Lan et al., 2020). For distillation, we evaluate *KD* (Hinton et al., 2015), *SHAKE* (Li & Zhe, 2022), and *C2KD* (Huo et al., 2024). All methods use the same pretrained encoders with trainable adapters. For distillation approaches, teacher and student adapters are unimodally pretrained with perturbation labels before fine-tuning via their respective methods. We benchmark PEA by applying it to $z_S$ during training, and compare it to existing biological and transcriptomics data augmentation approaches : MWO (Kircher et al., 2022), scVI denoising (Lopez et al., 2018), MDWGAN-GP (Li et al., 2023), scGFT (Nouri, 2025), and their combination with and without PEA. Hyperparameters are optimized via grid search on a validation split, with results averaged across multiple seeds.

**Evaluation Datasets.** We assess generalization on three Out-Of-Distribution (OOD) datasets, each introducing dis-

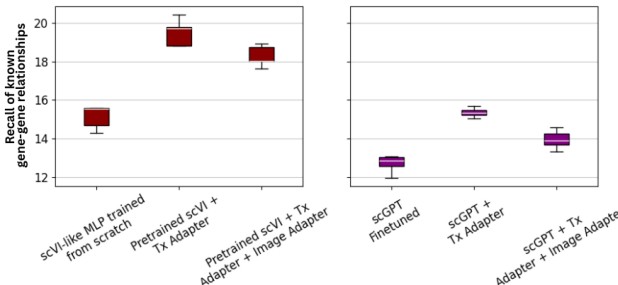

*Figure 1.* Impact of training choices on Semi-Clipped performance for known biological relationship recall on *HUVEC-KO*. Finetuning or multimodal training from scratch underperforms due to limited weakly paired data, while using adapters on pretrained models significantly improves results. The best performance is achieved with Semi-Clipped : a single transcriptomic adapter aligned to frozen image representations.

tinct distribution shifts. (1) **Experimental variability:** The *HUVEC-KO* dataset contains arrayed bulk transcriptomics data from 120,000 genetically perturbed sample, with around 300 CRISPR gene Knock-Out (KO) in HUVEC cells, unlike the training set, which uses chemical perturbations. This dataset does not share any experiment with the training set, and evaluates generalization to unseen experiments and unseen genetic perturbations. (2) **Quantification method shift:** The LINCS dataset (Subramanian et al., 2017) includes 443,000 arrayed bulk transcriptomics samples across 31 cell types and 5,157 CRISPR gene KO, using the L1000 assay, a transcript abundance measurement method different from the sequencing-based approach in training. (3) **Single-cell adaptation:** The *SC-RPE1* dataset (Replogle et al., 2022) consists of 247,914 single-cell transcriptomic samples from retinal pigmented epithelium cells with 2,393 CRISPR knockouts, testing the transition from bulk transcriptomics (training dataset) to single-cell transcriptomics. Together, these three OOD evaluation settings introduce significant distribution shifts on different aspects, testing the model's robustness to new cell types, experimental conditions, and gene expression quantification methods.

## 5. Results

We aim to evaluate the impact of Semi-Clipped and PEA both independently and in combination. Our primary objective is to improve biological relationship recall on OOD datasets compared to the corresponding unimodal transcriptomic baseline while preserving or enhancing interpretability in transcriptomics.

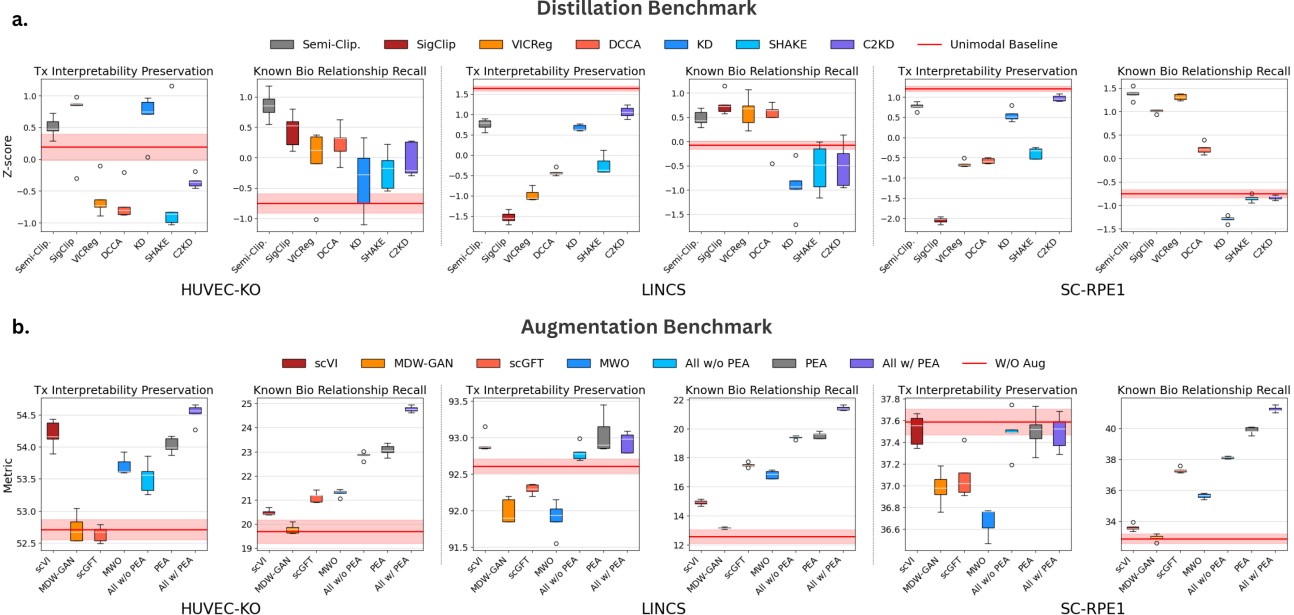

*Figure 2.* Performance comparison of the distillation and augmentation components of our approach compared to existing distillation methods (a) and biological data augmentation techniques (b) across five training seeds. Higher is better for all metrics. Semi-Clipped and PEA maintain interpretability and achieve the highest performance on all OOD datasets. (a) Z-scores of evaluation metrics (relationship recall and Tx preservability) are shown, with cool colors for label-based methods and warm colors for label-free approaches, without data augmentation. (b) Raw scores are shown for relationship recall and Tx preservability. Transcriptomics data augmentations, MWO (Kircher et al., 2022), scVI denoising (Lopez et al., 2018), MDWGAN-GP (Li et al., 2023), scGFT (Nouri, 2025), are applied **within Semi-Clipped training**. We compare training results where we simultaneously use all evaluated data augmentations, both with and without PEA, to assess its additional impact in a practical setting on both evaluation tasks.

## 5.1. Semi-Clipped Enables Robust and Generalizable Transcriptomic Representations

To analyze the effect of different training choices on Semi-Clipped performance, we first examine its impact on known biological relationship recall using the *HUVEC-KO* dataset. **Without data augmentations and using the CLIP loss**, we conduct two comparisons. Figure 1 (left) compares training an scVI-like MLP from scratch for the distillation task against using a pretrained scVI model. Additionally, it evaluates the effect of introducing an image adapter instead of relying solely on a transcriptomics adapter while keeping image embeddings frozen. Figure 1 (right) compares finetuning a pretrained scGPT model for distillation versus freezing scGPT and training a transcriptomics adapter on its own or with an image adapter. We find that leveraging a pretrained encoder with adapters consistently outperforms both training from scratch and finetuning for both scVI and scGPT. Furthermore, aligning transcriptomic representations to frozen image embeddings, as proposed in Semi-Clipped, yields superior performance compared to also training a microscopy imaging adapter.

We evaluate Semi-Clipped's ability to learn generalizable and biologically meaningful representations of transcriptomics compared to existing distillation methods, **using an**

**scVI pretrained encoder for transcriptomics**. For clarity, we define label-free approaches as those that do not use biological labels in the training objective, even if labels are used for modality pairing. To ensure a fair comparison of the core methods, **no data augmentation is applied**. Figure 2 (a) presents the performance of Semi-Clipped against various label-based and label-free distillation approaches on the Transcriptomic Interpretability Preservation and Known Biological Relationship Recall tasks across all three OOD datasets. Scores are standardized as z-scores and averaged over 5 seeds, with higher values indicating better performance. Distillation methods using label supervision (cool colors) generally show weaker relationship recall compared to unsupervised multimodal methods (warm colors) and even underperform the unimodal baseline in *LINCS* and *SC-RPE1*. In contrast, Semi-Clipped achieves the highest relationship recall in *HUVEC-KO* and *SC-RPE1* while also slightly surpassing the unimodal baseline in transcriptomics interpretability in *HUVEC-KO*. This suggests successful knowledge transfer from morphology to transcriptomics without sacrificing interpretability. In *LINCS*, Semi-Clipped performs competitively, outperforming all label-supervised distillation methods and the unimodal baseline in relationship recall while closely matching the best unsupervised multimodal methods. On the transcriptomics

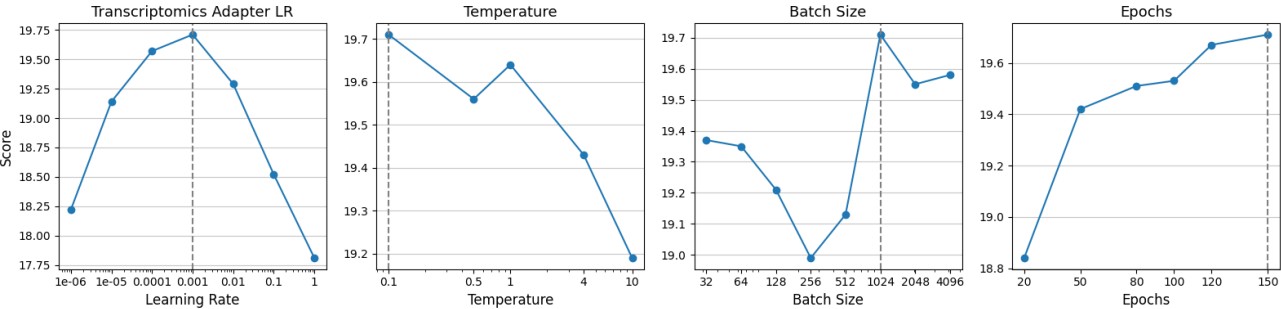

*Figure 3.* Ablation study on the known relationship recall score of hyperparameters choices (Tx Adapter learning rate, CLIP loss temperature, batch size, and training epochs) for training Semi-Clipped on the HUVEC-KO dataset, including the selected optimal configuration (dotted vertical line). For each studied parameter, we set all other hyperparameters at their best performing value. While performance varies with parameter changes, the method remains largely robust, showing minimal degradation and no collapse

preservation metric for *LINCS* and *SC-RPE1*, Semi-Clipped retains strong interpretability, slightly trailing the unimodal baseline but outperforming most distillation approaches. This minor limitation likely reflects the challenge of maintaining interpretability across unseen cell types. Overall, Semi-Clipped effectively balances generalization and interpretability across all OOD settings, consistently achieving the most robust performance across all metrics and demonstrating its strength as a distillation method.

To further assess Semi-Clipped's robustness, we conduct a detailed ablation study on individual hyperparameters when trained independently on the *HUVEC-KO* dataset (Figure 3). We isolate the effect of each parameter by fixing all others to their optimal values. The results reveal that while performance fluctuates with changes in configuration, Semi-Clipped remains resilient, exhibiting only minor degradation without any performance collapse. Optimal learning is achieved with a balanced learning rate, a lower temperature for the CLIP loss term, and larger batch sizes, though the method still performs competitively even with small batches. Additionally, increasing the number of training epochs yields substantial improvements. These findings reinforce the method's stability and reliability across a wide range of training conditions.

### 5.2. PEA Enhances Distillation Across Methods and Synergizes with Existing Augmentations

We further evaluate the effectiveness of our proposed PEA data augmentation in enhancing distillation performance across both evaluation tasks. **Using Semi-Clipped as the base model**, we compare its performance over five training seeds in three settings: (1) without any data augmentation, (2) with multiple existing biologically inspired transcriptomics augmentations from the literature, each used separately, and (3) with PEA as the sole augmentation. This initial evaluation isolates the specific contribution of PEA. Additionally, to reflect real-world training conditions where multiple augmentations are typically applied together, we

conduct a broader comparison. Specifically, we compare Semi-Clipped trained with all existing biological augmentations except PEA against its performance when trained with the full set of augmentations, including PEA. Figure 2 (b) presents the results of this comparison. Across all three evaluation datasets, PEA achieves state-of-the-art performance in Known Biological Relationship Recall, significantly outperforming all existing approaches. It also preserves transcriptomic interpretability, matching the no-augmentation baseline in *SC-RPE1* while surpassing it in *HUVEC-KO* and *LINCS*. Notably, PEA alone improves performance over not using augmentations by 17% in *HUVEC-KO*, 55% in *LINCS*, and 20% in *SC-RPE1*. More strikingly, PEA outperforms the combined effect of all other biological augmentations used together, highlighting its strong biological foundation and ability to introduce meaningful variation to the distillation process. Furthermore, integrating PEA with all other augmentations further enhances performance beyond using PEA alone, demonstrating its complementarity to existing transcriptomics augmentation techniques. This combined approach yields the highest overall improvements, increasing performance over the no-augmentation baseline by 25% in *HUVEC-KO*, 69% in *LINCS*, and 26% in *SC-RPE1*. These results confirm that PEA not only provides substantial individual benefits but also synergizes effectively with existing augmentation strategies.

We assess whether PEA enhances performance across different distillation approaches beyond Semi-Clipped and compare its impact on various methods. Specifically, we apply PEA to KD, SHAKE, VICReg, and Semi-Clipped and evaluate its effect on benchmark tasks. Each method is trained over 15 different seeds, both with and without PEA, and we use a Wilcoxon signed-rank test to determine the statistical significance of improvements. Table 1 summarizes the results: PEA consistently enhances performance across all three OOD datasets for every distillation approach, with particularly strong gains in *LINCS* and *SC-RPE1*. This confirms that PEA is broadly beneficial across methods. Notably,

| Method | HUVEC-KO | | LINCS | | SC-RPE1 | |
|---|---|---|---|---|---|---|
| | Tx Preservation | Known Relationships | Tx Preservation | Known Relationships | Tx Preservation | Known Relationships |
| Random baseline | $33.92 \pm 0.09$ | $10.37 \pm 0.11$ | $47.09 \pm 0.07$ | $10.81 \pm 0.04$ | $25.34 \pm 0.11$ | $10.03 \pm 0.02$ |
| Unimodal baseline | $52.23 \pm 0.34$ | $16.51 \pm 0.85$ | $\mathbf{93.35} \pm 0.07$ | $12.21 \pm 0.11$ | $\mathbf{37.75} \pm 0.28$ | $25.29 \pm 0.24$ |
| KD | $52.90 \pm 0.31$ | $16.00 \pm 1.36$ | $92.69 \pm 0.15$ | $11.83 \pm 0.27$ | $37.43 \pm 0.32$ | $23.9 \pm 0.18$ |
| KD + PEA | $\uparrow \mathbf{54.12} \pm 0.63$ | $\uparrow \underline{20.65} \pm 1.78$ | $\uparrow 93.11 \pm 0.29$ | $\uparrow 15.73 \pm 0.59$ | $\uparrow 37.55 \pm 0.38$ | $\uparrow 29.02 \pm 0.36$ |
| SHAKE | $51.93 \pm 0.80$ | $17.02 \pm 1.02$ | $91.64 \pm 0.23$ | $12.09 \pm 0.49$ | $36.95 \pm 0.46$ | $25.13 \pm 0.19$ |
| SHAKE + PEA | $\uparrow 52.93 \pm 0.83$ | $\uparrow 19.98 \pm 1.34$ | $\uparrow 92.43 \pm 0.31$ | $\uparrow 16.84 \pm 0.51$ | $\downarrow 36.15 \pm 0.51$ | $\uparrow 30.81 \pm 0.31$ |
| VICReg | $51.87 \pm 0.39$ | $17.25 \pm 1.14$ | $91.19 \pm 0.19$ | $12.96 \pm 0.45$ | $36.75 \pm 0.17$ | $32.19 \pm 0.26$ |
| VICReg + PEA | $\uparrow 53.76 \pm 0.66$ | $\uparrow 20.46 \pm 0.83$ | $\uparrow 91.22 \pm 0.25$ | $\uparrow \underline{18.12} \pm 0.19$ | $\downarrow 36.33 \pm 0.22$ | $\uparrow \underline{38.14} \pm 0.29$ |
| Semi-Clipped | $52.78 \pm 0.27$ | $19.71 \pm 1.19$ | $92.71 \pm 0.23$ | $12.68 \pm 0.33$ | $37.54 \pm 0.19$ | $32.65 \pm 0.21$ |
| Semi-Clipped + PEA | $\uparrow \underline{53.87} \pm 0.37$ | $\uparrow \mathbf{23.05} \pm 0.42$ | $\uparrow \underline{93.15} \pm 0.38$ | $\uparrow \mathbf{19.63} \pm 0.18$ | $\uparrow \underline{37.56} \pm 0.15$ | $\uparrow \mathbf{39.84} \pm 0.23$ |

*Table 1.* Performance improvement of different distillation methods with and without PEA under all OOD settings. We average the scores of 15 different seeds for each model, and the p-value of every result improvement is below 0.05 using the Wilcoxon signed-rank statistical test. Improvements from using PEA are indicated with upward arrows. For each OOD setting, the best-performing model is shown in bold, and the second-best is underlined. Using PEA as data augmentation for distillation approaches preserves the transcriptomics information while widely improving the zero-shot retrieval of known biological relationships for all the three OOD datasets used for evaluation.

it also significantly improves Transcriptomic Interpretability, likely due to its ability to preserve biological information while introducing controlled variations, enhancing the signal-to-noise ratio. All gains in Known Relationship Recall between PEA and non-PEA settings are statistically significant (p-values $< 0.05$). Importantly, Semi-Clipped remains the top-performing approach in Known Biological Relationship Recall across all evaluation datasets when using PEA, while also achieving the second-best performance in Transcriptomic Interpretability Preservation across all datasets.

We analyze the contribution of each PEA component to performance improvements by conducting an ablation study on the *HUVEC-KO* dataset. We evaluate its impact on KD, SHAKE, VICReg, and Semi-Clipped, progressively adding PEA components to the base distillation methods without augmentations. Each step in the ablation builds upon the previous one: (1) Fixed biological augmentation : applying a predefined set of batch correction techniques. (2) Inference on TVN-corrected embeddings : applying Typical Variation Normalization (TVN) (Ando et al., 2017) correction to $z_S$ at inference before passing them to the adapter $f_S$. (3) Augmentation stochasticity : randomly dropping a subset of batch correction steps to introduce variation. (4) Control sampling : randomly sampling a varying amount of control samples for correction, completing the full PEA approach. Table 2 summarizes the results, averaged over 15 seeds and reporting Known Biological Relationship Recall for each distillation approach. Every component contributes incremental improvements, with control sampling providing the strongest boost, particularly for Semi-Clipped. Importantly, all distillation methods show consistent performance gains at each step, indicating that each PEA component plays a critical role in enhancing distillation outcomes.

## 5.3. Semi-Clipped with PEA Enables Synergistic Integration of Morphological and Transcriptomic Insights

We analyze the biological insights provided by Semi-Clipped trained with PEA, comparing the known biological relationships it retrieves to those identified independently by unimodal microscopy imaging and transcriptomics models on the *HUVEC-KO* OOD dataset. Specifically, we evaluate the quantity and overlap of relationships retrieved by KD, SHAKE, VICReg, and Semi-Clipped, all trained with PEA, to assess whether these models remain faithful to transcriptomics-specific relationships or exhibit modality drift. This is quantified by measuring the intersection between relationships retrieved by each distillation method and those identified by the unimodal transcriptomics model. Figure 4 presents Venn diagrams of these intersections. Semi-Clipped shows strong alignment with transcriptomics-retrieved relationships while also capturing additional biological insights typically associated with morphological features. In contrast, while the other distillation approaches retrieve many known relationships, they exhibit minimal overlap with those identified by transcriptomics alone. Notably, KD and SHAKE, both label-based methods, demonstrate particularly weak alignment with transcriptomics relationships, likely due to the confounding effects of weak biological labels used during training. These findings suggest that Semi-Clipped effectively preserves transcriptomic insights while significantly enriching them with complementary morphological information, achieving a better balance between biological faithfulness and multimodal integration.

We next investigate whether distilling morphological features into transcriptomics yields a purely additive effect or if it generates emergent synergies between modalities. To assess this, we analyze the set *Distillation \ (Transcriptomics ∪ Microscopy)* in Figure 4, representing relationships

| PEA Configuration | KD | SHAKE | VICReg | Ours |
|---|---|---|---|---|
| Base Method | 16.00 | 17.02 | 17.25 | 19.71 |
| + Fixed Bio-Aug | 16.76 | 17.22 | 17.97 | 19.95 |
| + Inference on TVN | 18.76 | 18.32 | 18.62 | 20.58 |
| + Aug. Stochasticity | 19.37 | 18.84 | 19.43 | 21.79 |
| + Ctrl Sampling (PEA) | **20.65** | **19.98** | **20.46** | **23.05** |

*Table 2.* Ablation study of different PEA components on *HUVEC-KO* evaluation dataset, on retrieval of known relationships of different distillation methods, averaged over 15 seeds. We see that combining all PEA components achieves significant improvements, especially when using our Semi-Clipped approach.

uniquely retrieved by the distillation model but absent in unimodal transcriptomics or microscopy imaging. For all evaluated methods, we perform Gene-Set Enrichment Analysis (GSEA) (Subramanian et al., 2005) to identify enriched biological pathways within this set compared to other distinct relationships retrieved by each distillation approach, filtering for gene sets with p-values $< 0.01$. Surprisingly, KD, SHAKE, and VICReg fail to significantly enrich any biological pathway, whereas Semi-Clipped uniquely enriches pathways related to the cell cycle and post-translational modifications (Appendix Table 3). This suggests that distilling morphological traits into transcriptomics using our approach enhances the capture of cell cycle-related information, which may be less detectable or noisier in either modality alone. This outcome likely arises from Semi-Clipped's ability to integrate rich phenotypic information from microscopy imaging, including morphological traits, spatial organization, and cellular process indicators, with transcriptomic markers such as mitochondrial RNA gene counts, often associated with cell cycle activity. This fusion enables a deeper biological synergy, allowing distillation to reveal novel biological insights that neither modality could achieve independently, while still permitting unimodal inference rather than requiring multimodal fusion.

## 6. Discussion

In this work, we introduced Semi-Clipped, a self-supervised framework for distilling morphological knowledge of biology into transcriptomic representations using multimodal alignment techniques. Additionally, we proposed PEA, a biologically informed augmentation strategy that repurposes batch correction to enhance representation learning. Our results demonstrate that Semi-Clipped outperforms existing distillation methods while preserving transcriptomic interpretability. Furthermore, we show that label-free distillation consistently surpasses label-based approaches, reinforcing that biological labels often lack the granularity needed to fully capture cellular complexity. A key contribution of this work is the reinterpretation of batch correction as a biologically meaningful data augmentation. Unlike conventional transcriptomic data augmentations that may disrupt

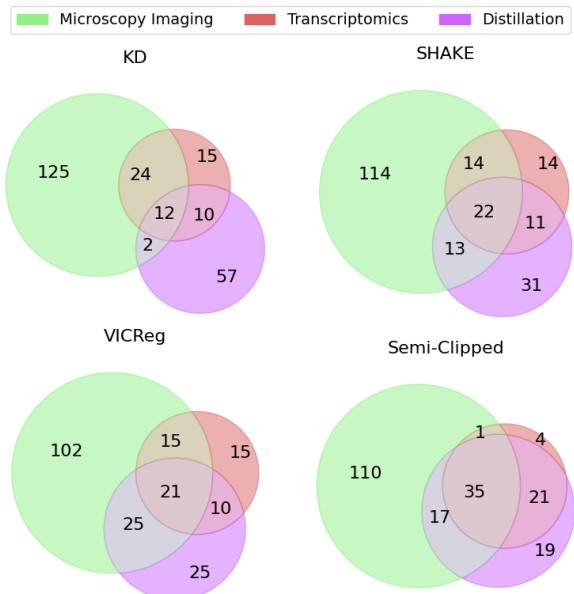

*Figure 4.* Venn diagrams of retrieved biological relationships for KD, SHAKE, VICReg, and Semi-Clipped (all trained with PEA) on the *HUVEC-KO* OOD dataset. Semi-Clipped shows the highest overlap with transcriptomics while integrating morphological insights, whereas KD and SHAKE exhibit the weakest alignment, possibly due to reliance on weak biological labels. Detailed measures of the gains and losses of each method in each modality are available in Figure 5.

critical expression signals, PEA introduces plausible variability while maintaining essential biological properties. This approach significantly improves cross-modal distillation performance, increasing Known Biological Relationship Recall in OOD tests while preserving interpretability. Beyond aligning transcriptomic and morphological information, Semi-Clipped reveals emergent biological synergies, particularly in cell cycle regulation and post-translational modifications. Despite these advantages, challenges remain. Random pairing within treatment groups may dilute representation quality when subtle intra-group differences exist, highlighting the need for better matching strategies. Limited large-scale paired data also restricts broader applicability. Nonetheless, Semi-Clipped is computationally efficient: training on a scaled version of our dataset with 1.3 million weakly paired samples takes only 19 hours on a single H100 GPU, thanks to the use of frozen backbones and lightweight adapters. As multimodal datasets grow, scaling these methods could further advance biological research and cross-modal understanding.

## Impact Statement

This paper presents work whose goal is to advance the field of Machine Learning in its application to life sciences. There are many potential societal consequences of our work,

especially relating to the discovery of new biological relationships and potential drug treatments. Utmost care should be taken to validate safety and efficacy of model predictions in pre-clinical trials.

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

# A. Detailed Experimental Setup

## A.1. Encoders

We use three main models for our experiments, in addition to the MLP trained from scratch. Phenom-1 (Kraus et al., 2024) is a Vision Transformer-based model with 300 million parameters, trained using a Masked Autoencoder (MAE) framework. It is pretrained on RPI-93M, a dataset of 93 million microscopy images, capturing diverse cellular phenotypes across CRISPR, chemical, and soluble perturbations, making it highly effective for large-scale cellular morphology analysis. scVI (Lopez et al., 2018) is a probabilistic generative model designed for single-cell RNA sequencing (scRNA-seq) data, trained using a Variational Autoencoder (VAE) framework. It encodes high-dimensional gene expression data into a biologically meaningful latent space, leveraging a zero-inflated negative binomial (ZINB) reconstruction objective to model overdispersion and dropout effects in transcriptomic data. scGPT (Cui et al., 2024) is a transformer-based foundation model pretrained on 33 million scRNA-seq samples using a masked language modeling objective. It captures complex gene–gene and gene–cell interactions, with fine-tuning capabilities for tasks like cell type annotation, multi-omic integration, and perturbation response prediction. These models provide robust, biologically relevant representations tailored for microscopy and transcriptomics data.

## A.2. Implementation details

The MLP adapters $f_S$ and $f_T$ used in this work are fully connected feedforward networks designed to align embeddings from the transcriptomics (Tx) and microscopy imaging encoders into a shared latent space. For the transcriptomics adapter, the architecture comprises an input layer of size 256, two hidden layers with dimensions 512 and 1024 respectively, and an output layer of size 768. The image adapter follows a similar design, with an input size of 768, two hidden layers of size 1024, and an output layer of size 768. ReLU activations are applied to all hidden layers, while the output layer uses a linear activation.

For VICReg, learning rates for the Tx and image adapters were 0.1 and $1 \times 10^{-8}$, respectively, and training spanned 10 epochs with a minimum learning rate of $10^{-10}$; the VICReg loss parameters (similarity, variance, and covariance weights) were kept at their default settings. Similarly, SigClip used Tx and Img adapters learning rates of 0.1 and $10^{-8}$, respectively, with training conducted for 10 epochs and a minimum learning rate of $10^{-10}$; the temperature and normalization parameters were kept at their defaults. For DCCA, the Tx and Img adapters were trained with a learning rate of $10^{-6}$ and $10^{-8}$ respectively over 50 epochs, a minimum learning rate of $10^{-10}$, and loss parameters including an output dimension size of 30, usage of all singular values, and an epsilon of $10^{-6}$. The SHAKE method utilized Tx and Img adapters learning rates of 0.1 and $10^{-8}$, Tx and Img classifier learning rates of $10^{-4}$ and $10^{-7}$, respectively, and a temperature of 9, with loss balancing hyperparameters $\alpha = 10$ and $\beta = 0.001$; training was conducted over 10 epochs with a minimum learning rate of $10^{-10}$. For KD, the Tx adapters and classifier learning rates were 0.1 and $10^{-4}$, respectively, with a temperature of 9 and $\alpha = 10$, trained for 10 epochs with a minimum learning rate of $10^{-10}$. Lastly, C2KD employed Tx and Img adapters learning rates of 0.1 and $10^{-6}$, Tx and Img classifier learning rates of $10^{-5}$ and $10^{-3}$, respectively, and a temperature of 2 with a Kendall Rank Correlation threshold of 0.3; training spanned 30 epochs with a minimum learning rate of $10^{-7}$. At evaluation step, we perform TVN alignment for all output embeddings for the Known Relationship Recall benchmark, and use raw embeddings for Transcriptomic Interpretability Preservation benchmark, as is used in (Bendidi et al., 2024b).

# B. Evaluation tasks

## B.1. Known Biological Relationship Recall

The *Known Relationship Recall* score is a benchmarking metric introduced in (Celik et al., 2024) and designed to evaluate the extent to which a perturbative map captures established biological relationships. This score serves as a proxy for assessing the biological relevance of the map and its ability to uncover meaningful interactions between genes. By comparing predicted relationships within the map to curated annotations from biological databases, the Known Relationship Recall score provides a quantitative measure of the map's fidelity to known biology.

The computation of the Known Relationship Recall score follows these steps:

**1. Pairwise Similarity Computation:** For each pair of genes $(g_i, g_j)$ in the map, we compute the cosine similarity between

their aggregated embeddings $\mathbf{x}_{g_i}$ and $\mathbf{x}_{g_j}$. The cosine similarity is defined as:

$$\cos(\mathbf{x}_{g_i}, \mathbf{x}_{g_j}) = \frac{\langle \mathbf{x}_{g_i}, \mathbf{x}_{g_j} \rangle}{\|\mathbf{x}_{g_i}\| \, \|\mathbf{x}_{g_j}\|},$$

where $\langle \mathbf{x}_{g_i}, \mathbf{x}_{g_j} \rangle$ is the dot product of the embeddings, and $\|\mathbf{x}_{g_i}\|$ is the Euclidean norm of $\mathbf{x}_{g_i}$.

**2. Selection of Predicted Relationships:** Relationships are classified as "predicted" if their cosine similarity scores fall into the top or bottom relationships according to a percentage threshold (usually 5%) of the distribution of all pairwise similarities. High similarity scores indicate cooperative relationships, while low scores suggest functional opposition.

**3. Validation Against Biological Databases:** The predicted relationships are validated using established biological annotations from databases such as CORUM, HuMAP, Reactome, SIGNOR, and StringDB. Only gene that appear in the perturbation dataset are considered for pairs in the database.

**4. Recall Calculation for Each Database:** For each database, the recall is computed as the fraction of annotated relationships that are successfully identified among the predicted relationships:

$$\text{Recall}_{\text{db}} = \frac{\#(\text{True Positive Relationships})}{\#(\text{Total Annotated Relationships in Map})}$$

Here, true positive relationships are those annotated in the database that also fall within the predicted set. The final Known Relationship Recall score is computed as the mean of the recall values across the five databases.

The Known Relationship Recall score provides a single aggregated metric that encapsulates the map's ability to recapitulate established biological relationships. A high score indicates strong alignment with existing annotations, demonstrating the map's utility in representing meaningful biological interactions.

### B.2. Transcriptomic Interpretability Preservation

The *Transcriptomic Interpretability Preservation*, first introduced as linear interpretability evaluation in (Bendidi et al., 2024b), is an evaluation framework designed to assess how well a model captures and preserves biologically meaningful patterns in transcriptomic data. This task evaluates the quality of the model's internal representations and their ability to reconstruct gene expression profiles accurately while maintaining the structural relationships between control and perturbation conditions. By focusing on both the accuracy of reconstructed gene expression profiles and the preservation of batch-specific control-perturbation relationships, this metric provides a holistic view of the model's capability to retain original transcriptomic interpretability. The evaluation relies on two complementary metrics, which are averaged to compute the final Transcriptomic Interpretability Preservation score:

**Structural Integrity Score:** This metric quantifies how well the model preserves the relationships between control and perturbation conditions within each biological batch. The Structural Integrity score is computed as:

$$\text{Structural Integrity} = 1 - \frac{\text{Structural Distance}}{\text{Structural Distance}_{\text{max}}},$$

where the Structural Distance measures the Frobenius norm of the difference between centered predicted and actual gene expression matrices, and Structural Distance$_{\text{max}}$ is the theoretical maximum distance, as derived in (Bendidi et al., 2024b). A score close to 1 indicates strong preservation of the structural relationships.

**Spearman Correlation of Reconstruction:** This metric evaluates how accurately the model reconstructs original gene expression profiles from its internal latent representations. The Spearman correlation is calculated between the predicted and true gene expression profiles, providing a robust measure of rank-based agreement.

To provide a comprehensive evaluation, the *Transcriptomic Interpretability Preservation* metric is computed as the average of the Structural Integrity score and the Spearman correlation of reconstruction. By evaluating both aspects, the metric ensures that a model not only produces high-quality reconstructions but also retains the underlying biological structure of the data. This is crucial for downstream applications such as identifying gene interactions or studying the effects of perturbations in various conditions.

# C. Batch Correction Techniques

## C.1. Centering

Centering involves adjusting the dataset such that each feature has a mean of zero. This is achieved by subtracting the mean of each feature from the data. Given a feature matrix $X \in \mathbb{R}^{n \times m}$, where $n$ is the number of samples and $m$ is the number of features, the centered matrix $\tilde{X}$ is computed as:

$$\tilde{X}_{ij} = X_{ij} - \frac{1}{n} \sum_{k=1}^{n} X_{kj}, \quad \forall i = 1, \ldots, n, \quad \forall j = 1, \ldots, m.$$

This step shifts the data so that each feature's mean is zero. In batch-corrected biological datasets, centering is typically applied to remove the influence of negative control embeddings, facilitating the focus on perturbation effects.

## C.2. Center Scaling/Standardization

Center scaling/Standardization extends centering by adjusting each feature so that it has unit variance. This ensures comparability across features. For a centered matrix $\tilde{X}$, the scaled matrix $\hat{X}$ is defined as:

$$\hat{X}_{ij} = \frac{\tilde{X}_{ij}}{\sigma_j}, \quad \sigma_j = \sqrt{\frac{1}{n} \sum_{k=1}^{n} \tilde{X}_{kj}^2},$$

$$\forall i = 1, \ldots, n, \quad \forall j = 1, \ldots, m,$$

where $\sigma_j$ represents the standard deviation of the $j$-th feature. Center scaling is important for techniques like Principal Component Analysis (PCA), which are influenced by the scale of the data.

## C.3. Typical Variation Normalization (TVN)

Typical Variation Normalization (TVN) is a technique designed to enhance the representation of biological data by minimizing batch effects and accentuating subtle phenotypic differences. TVN is particularly relevant in high-content imaging screens and other scenarios with significant batch variability.

TVN begins by computing the principal components of control samples (negative control conditions) to identify the primary directions of variation. PCA is performed on the centered control data $\tilde{X}_{\text{control}}$ to obtain principal components $\{\mathbf{v}_1, \ldots, \mathbf{v}_m\}$, with each component representing a variance direction in the data space. The normalization process involves the following steps:

**1. Centering and Scaling of Negative Controls:** The negative control data $X_{\text{control}}$ is centered and scaled as:

$$\hat{X}_{\text{control}} = \frac{X_{\text{control}} - \mu_{\text{control}}}{\sigma_{\text{control}}},$$

where $\mu_{\text{control}}$ and $\sigma_{\text{control}}$ are the mean and standard deviation of the control embeddings.

**2. Principal Component Analysis (PCA):** PCA is conducted on $\hat{X}_{\text{control}}$ to derive principal components. The matrix $W \in \mathbb{R}^{m \times m}$ consists of columns that are the component vectors $\mathbf{v}_j$.

**3. TVN Transformation:** The transformation matrix $T$ is constructed to normalize variance along each principal component axis:

$$T = W \cdot D^{-1/2} \cdot W^\top,$$

where $D$ is a diagonal matrix of the eigenvalues associated with the principal components.

**4. Application to All Embeddings:** The transformation is applied to all embeddings $X_{\text{all}}$ as:

$$X_{\text{TVN}} = T \cdot X_{\text{all}}.$$

This step reduces unwanted variation while emphasizing important biological differences, enabling a focus on subtle or rare phenotypic features without batch-related artifacts.

# D. Additional Results

| # of relationship counts | KD | SHAKE | VICReg | Semi-Clipped |
|---|---|---|---|---|
| Distillation Relationship Gains relative to Tx | 59 | 44 | 50 | 36 |
| Distillation Relationship Losses relative to Tx | 39 | 28 | 30 | 5 |
| Transcriptomic Relationship Preservation | 22 | 33 | 32 | 56 |
| Total Recalled Relationships | 81 | 77 | 82 | 92 |

*Figure 5.* Comparison of relationship gains and losses across cross-modal distillation methods shown in Figure 4. Our approach achieves the highest overall relationship recall and best preserves transcriptomic information.

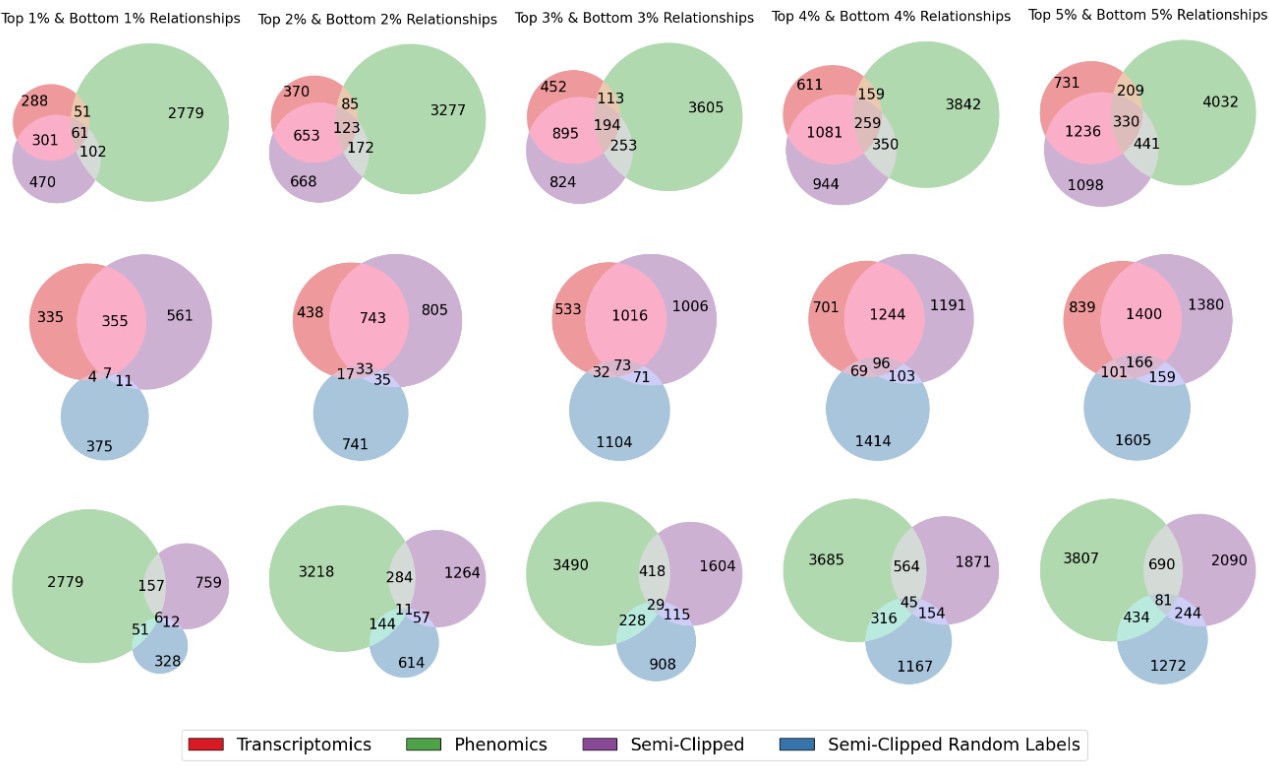

*Figure 6.* Literature-known biological relationships retrieved through the *LINCS* dataset by the transcriptomics and microscopy imaging unimodal encoders, alongside our proposed Semi-Clipped approach, without data augmentations, and a null distribution through randomization of the perturbation labels of the pretrained Semi-Clipped. Semi-Clipped remains consistent with transcriptomics while distilling new relationships from microscopy imaging, displaying a distinct pattern from the null distribution.

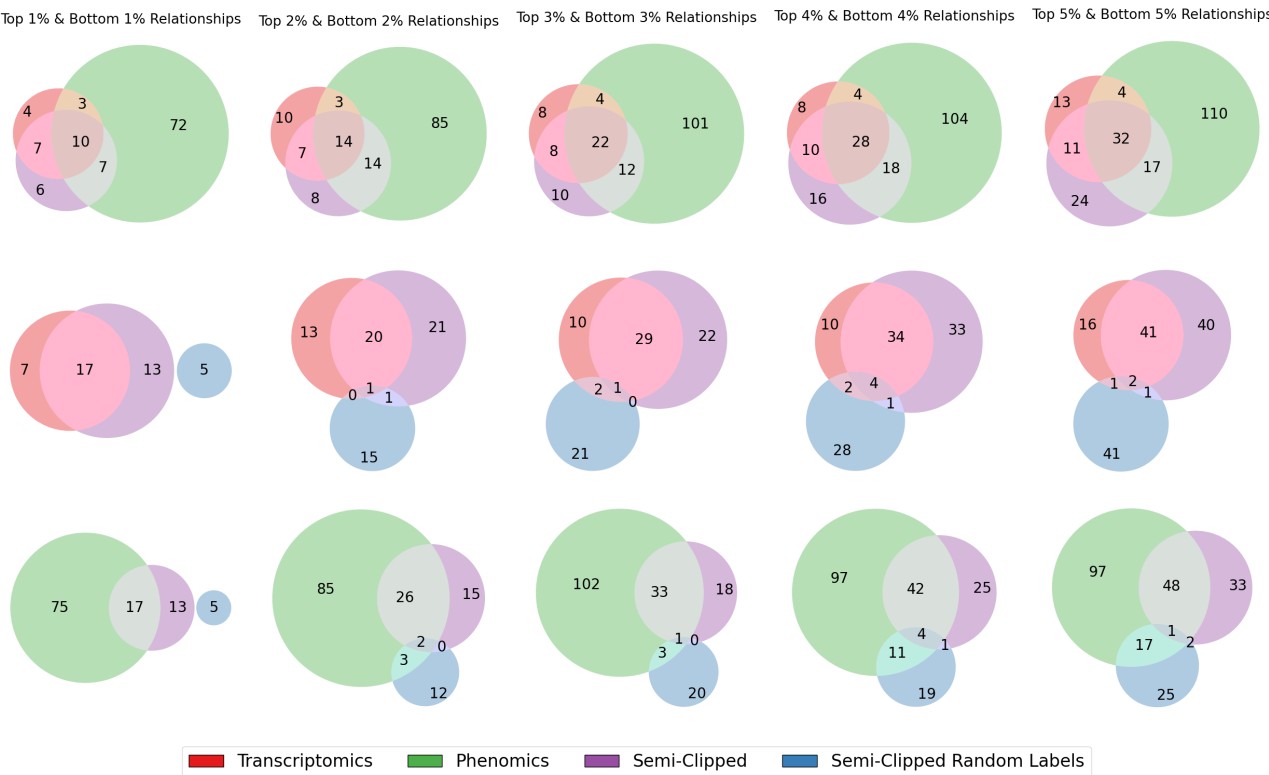

*Figure 7.* Literature-known biological relationships retrieved by the transcriptomics and microscopy imaging unimodal encoders, alongside our proposed Semi-Clipped approach (first row), without data augmentations, across different retrieval thresholds (columns) on the *HUVEC-KO* dataset. Semi-Clipped remains consistent with transcriptomics while distilling new relationships from microscopy imaging. In second and third row, we compare Semi-Clipped to a null distribution achieved through randomization of the perturbation labels of the pretrained Semi-Clipped. Our approach displays a distinct pattern from the null distribution, and aligns better to both modalities than random.

| Enriched Pathways | Source Set | P-value |
|---|---|---|
| REACTOME CELL CYCLE CHECKPOINTS | Semi-Clipped \ (Tx ∪ Img) | 0.0323 |
| KEGG ANTIGEN PROCESSING AND PRESENTATION | (Semi-Clipped ∩ Img) \ Tx | 0.0049 |
| KEGG P53 SIGNALING PATHWAY | (Semi-Clipped ∩ Img) \ Tx | 0.0033 |
| KEGG RIG I LIKE RECEPTOR SIGNALING PATHWAY | (Semi-Clipped ∩ Img) \ Tx | 0.0082 |
| REACTOME ADAPTIVE IMMUNE SYSTEM | (Semi-Clipped ∩ Img) \ Tx | 0.0114 |
| REACTOME ANTIGEN PRESENTATION FOLDING ASSEMBLY AND PEPTIDE LOADING OF CLASS I MHC | (Semi-Clipped ∩ Img) \ Tx | 0.0049 |
| REACTOME ANTIGEN PROCESSING CROSS PRESENTATION | (Semi-Clipped ∩ Img) \ Tx | 0.0016 |
| REACTOME ASPARAGINE N LINKED GLYCOSYLATION | (Semi-Clipped ∩ Img) \ Tx | 0.0049 |
| REACTOME CALNEXIN CALRETICULIN CYCLE | (Semi-Clipped ∩ Img) \ Tx | 0.0049 |
| REACTOME CELL CYCLE | (Semi-Clipped ∩ Img) \ Tx | 0.0480 |
| REACTOME CLASS I MHC MEDIATED ANTIGEN PROCESSING PRESENTATION | (Semi-Clipped ∩ Img) \ Tx | 0.0049 |
| REACTOME DDX58 IFIH1 MEDIATED INDUCTION OF INTERFERON ALPHA BETA | (Semi-Clipped ∩ Img) \ Tx | 0.0082 |
| REACTOME DEUBIQUITINATION | (Semi-Clipped ∩ Img) \ Tx | 0.0130 |
| REACTOME G1 S DNA DAMAGE CHECKPOINTS | (Semi-Clipped ∩ Img) \ Tx | 0.0033 |
| REACTOME G ALPHA Q SIGNALLING EVENTS | (Semi-Clipped ∩ Img) \ Tx | 0.0065 |
| REACTOME HEMOSTASIS | (Semi-Clipped ∩ Img) \ Tx | 0.0082 |
| REACTOME INNATE IMMUNE SYSTEM | (Semi-Clipped ∩ Img) \ Tx | 0.0227 |
| REACTOME NEGATIVE REGULATORS OF DDX58 IFIH1 SIGNALING | (Semi-Clipped ∩ Img) \ Tx | 0.0033 |
| REACTOME N GLYCAN TRIMMING IN THE ER AND CALNEXIN CALRETICULIN CYCLE | (Semi-Clipped ∩ Img) \ Tx | 0.0049 |
| REACTOME OVARIAN TUMOR DOMAIN PROTEASES | (Semi-Clipped ∩ Img) \ Tx | 0.0016 |
| REACTOME PLATELET ACTIVATION SIGNALING AND AGGREGATION | (Semi-Clipped ∩ Img) \ Tx | 0.0065 |
| REACTOME POST TRANSLATIONAL PROTEIN MODIFICATION | (Semi-Clipped ∩ Img) \ Tx | 0.0002 |
| REACTOME REGULATION OF TP53 ACTIVITY | (Semi-Clipped ∩ Img) \ Tx | 0.0179 |
| REACTOME REGULATION OF TP53 ACTIVITY THROUGH METHYLATION | (Semi-Clipped ∩ Img) \ Tx | 0.0016 |
| REACTOME REGULATION OF TP53 ACTIVITY THROUGH PHOSPHORYLATION | (Semi-Clipped ∩ Img) \ Tx | 0.0179 |
| REACTOME REGULATION OF TP53 EXPRESSION AND DEGRADATION | (Semi-Clipped ∩ Img) \ Tx | 0.0016 |
| REACTOME RNA POLYMERASE II TRANSCRIPTION | (Semi-Clipped ∩ Img) \ Tx | 0.0480 |
| REACTOME SIGNALING BY GPCR | (Semi-Clipped ∩ Img) \ Tx | 0.0082 |
| REACTOME STABILIZATION OF P53 | (Semi-Clipped ∩ Img) \ Tx | 0.0016 |
| REACTOME TRANSCRIPTIONAL REGULATION BY TP53 | (Semi-Clipped ∩ Img) \ Tx | 0.0195 |

*Table 3.* Gene Set Enrichment Analysis (GSEA) results on the *HUVEC-KO* dataset, highlighting enriched pathways identified uniquely in the Semi-Clipped approach compared to the transcriptomics and microscopy imaging unimodal encoders. The first row represents pathways uniquely enriched in Semi-Clipped after excluding the union of transcriptomics and morphological relationships, revealing enrichment in cell cycle pathways. The subsequent rows list pathways enriched in the intersection of Semi-Clipped and microscopy imaging, excluding transcriptomics relationships, which shows that in addition to Semi-Clipped unique enriched pathways, our approach is also enriched by morphology specific pathways.

