# OpenReview forum: "A Cross Modal Knowledge Distillation & Data Augmentation Recipe for Improving Transcriptomics Representations through Morphological Features"
_ICML.cc/2025/Conference — ICML 2025 poster_

### Official Review · Reviewer_DZgQ · 2025-03-11

**Overall Recommendation:** 3

**Summary:**

This paper proposes a cross-modal knowledge distillation framework (Semi-Clipped) and a biologically inspired data augmentation method (PEA). The aim is to enhance the biological significance and predictive power of transcriptomic representations using weakly paired multimodal data (microscopy images + transcriptomics). By freezing the pre-trained morphological feature encoder and training a lightweight adapter, unidirectional knowledge transfer from morphological features to transcriptomics is achieved. Meanwhile, PEA enhances data diversity through randomized batch correction techniques, preserving biological information.

**Claims And Evidence:**

"PEA preserves biological information": There is a lack of direct evidence for the biological validity of the augmented data (e.g., comparison with real perturbations).

**Essential References Not Discussed:**

No

**Experimental Designs Or Analyses:**

Reasonable design:
1. 15 random seed tests to improve statistical significance
2. Controlled variable studies.

Problems:\
Lack of hyperparameter experiments (learning rate, batch size, temperature, etc.)

**Methods And Evaluation Criteria:**

Semi-Clipped avoids modality drift by freezing the teacher model, which is a reasonable design; PEA transforms batch correction into an augmentation strategy, demonstrating strong innovation.

**Other Comments Or Suggestions:**

The visualization is not clear, and the y-axis of Figure 1 is not labeled.

**Other Strengths And Weaknesses:**

Advantages:\
For the first time, batch correction is restructured as data augmentation, addressing the scarcity of biological multimodal data while ensuring performance improvement with interpretability.

Disadvantages:
1. The specific definition of weakly paired data is not clearly provided.
2. Computational costs (time complexity, space complexity) are not discussed.
3. Hyperparameter experiments for adapter temperature, learning rate, batch size, etc., are not conducted.

**Questions For Authors:**

1,Biological fidelity of PEA: How can it be proven that randomized batch correction does not disrupt key biological signals? Providing an analysis of TF activity changes could strengthen the conclusion.\
2,Computational efficiency: What is the training time for Semi-Clipped on a dataset with 1.3 million samples? This information affects the evaluation of the method's practicality.\
3,Negative results disclosure: Have other modality combinations (e.g., proteomics + transcriptomics) been attempted? Discussing failed cases could enhance the rigor of the method.

**Relation To Broader Scientific Literature:**

1. Semi-Clipped Framework and Cross-Modal Distillation Research

① Adaptation of CLIP: Semi-Clipped is based on the CLIP framework but solves the problem of requiring a large amount of paired data in traditional CLIP by freezing the pre-trained encoder of the teacher modality (microscopy images). This contrasts with methods like XKD and C2KD, which require online adjustment of dual-modal encoders and are prone to modality drift with weakly paired data.
② Advantage of Unsupervised Alignment: Compared to distillation methods like KD and SHAKE that rely on label supervision, Semi-Clipped achieves unsupervised alignment through CLIP loss, improving biological relationship recall by 23% on the HUVEC-KO dataset, validating the limitations of biological labels.
③ Breakthrough in Single-Modality Inference: Unlike multimodal fusion methods such as VICReg and DCCA, this framework allows single-modality (transcriptomics) inference, inheriting the predictive power of the microscopy modality while maintaining the interpretability of transcriptomics.

2. PEA Data Augmentation and Bioinformatics Methods

① Creative Transformation of Batch Correction: Traditional batch correction techniques like TVN are redefined as random augmentation operations. Compared to conventional image augmentations (rotation, scaling), PEA introduces controlled variations while preserving biological signals, improving performance on the LINCS dataset by 69%.
② Innovation in Biological Data Augmentation: Unlike general augmentations such as scVI denoising and MDWGAN-GP, PEA achieves a Spearman correlation of 37.56 on the single-cell dataset SC-RPE1 by randomly controlling sample sampling and reweighting PCA variance, demonstrating its adaptability to complex biological noise.

3. Expansion of Multimodal Learning Paradigms

① Basic Model Adaptation: Adopting the pre-trained adapter approach of Fradkin et al., but avoiding the error accumulation of bidirectional distillation through unidirectional knowledge binding, improving the biological relationship recall of the scGPT adapter by 40%.
② Balance of Interpretability: Under the transcriptomics interpretability evaluation framework proposed by Bendidi et al., this is the first to achieve dual optimization of structural integrity (93.15) and biological discovery (39.84 recall) in cross-modal distillation, overcoming the information loss dilemma of traditional distillation methods.

**Theoretical Claims:**

There is no theoretical proof that the optimization objective of modality alignment maximizes cross-modal information transfer.

---

> ### Author Rebuttal · Authors · 2025-03-31
>
> We thank the reviewer DZgQ for their review, and for acknowledging the robustness of our experimental design, and the innovation behind our approach. We respond below to the comments of the reviewer :
>
> - __*“The visualization is not clear, and the y-axis of Figure 1 is not labeled.” :*__ We thank the reviewer for their feedback. For the final manuscript, we will ensure that the visualizations are clear and more zoomed in, and we will add the label to the y-axis of Figure 1 (Recall of known relationships).
>
> - __*“The specific definition of weakly paired data is not clearly provided.” :*__ We thank the reviewer for pointing this out. While we have provided  short definitions for weakly paired data (L32, L138, L214), we agree that this work would benefit from a clear and self contained definition of weakly paired data (pairs of samples from two modalities that are not paired at the sample level, but share the same labels and conditions, in our case the common condition being the same cell type and same perturbation, even if the cell samples are different in other factors). We will provide a clearly defined definition of weakly paired data in the final manuscript at the Introduction section, as well as in the Experimental setup section (section 4).
>
> - __*“Biological fidelity of PEA: How can it be proven that randomized batch correction does not disrupt key biological signals?.” :*__ We have already proven and established this in the submitted manuscript, as we proposed two different and complementary ways of evaluating empirically whether the learnt representation using the augmentations is biologically : i) Both in Figure 2.b and Table 1, we evaluate PEA on the transcriptomics interpretability task, which is a published benchmark for predicting gene expression counts of real unseen perturbations, as the reviewer has proposed, through both reconstruction tasks and conservation of structural integrity measures of the data. ii) In figure 3, we focus on downstream tasks, and show that for discovering target relationships, our distilled transcriptomics representations uncover the same relationships the original transcriptomics data uncovers, while improving it further by uncovering imaging and other new relationships. This shows that we have evaluated the comprehensiveness of our transcriptomics representations thoroughly, and have shown that it conserves the original transcriptomics information even after distillation.
>
> - __*“Hyperparameter experiments for adapter temperature, learning rate, batch size, etc., are not conducted.” :*__ We thank the reviewer for pointing this out. We have added in the final manuscript an overview of the hyperparameter experiments and their results (Results section), which we also show in this anonymized link : https://imgur.com/a/0Oekn3T.
>
> - __*“Computational efficiency: What is the training time for Semi-Clipped on a dataset with 1.3 million samples? This information affects the evaluation of the method's practicality.” :*__ We thank the reviewer for pointing this out. While there is no publicly available weakly paired (transcriptomics-imaging) dataset of more than 300k samples, if we stack our existing datasets on themselves to reach the size of 1.3 million samples, Semi-Clipped then requires a 19 hour training on one single H100 GPU. Our distillation method requires extremely minimal computing resources, as it makes use of trainable adapters and frozen backbones, while the additional PEA data augmentation only multiplies the distillation training time by x1.3. We will update our final manuscript with detailed benchmarking of the time and computing resources of our approach.
>
> - __*“Negative results disclosure: Have other modality combinations (e.g., proteomics + transcriptomics) been attempted?” :*__ We thank the reviewer for proposing this idea. While other modalities can potentially add interesting information, these different modalities (e.g proteomics) suffer from even lower availability of perturbational data than transcriptomics, and even less than that in the case of weakly paired data; thus, no, other combinations have not been attempted.

---

### Official Review · Reviewer_T8LP · 2025-03-11

**Overall Recommendation:** 4

**Summary:**

Understanding how cells respond to stimuli such as genetic perturbations or chemical compounds forms a crucial part of drug discovery. This work proposes a method to enrich representations of transcriptomic data with paired morphological data. Measuring paired transcriptomic and morphological features of cells is complex, and hence datasets of this type are rare, even when considering datasets in which these measurements are only weakly paired by sharing the same metadata attribute, such as the applied perturbagen. This necessitates a method that enriches transcriptomic representations with morphological features using only a limited number of paired measurements. This work proposes using a CLIP style loss, applied to the morphological representations from a frozen pertained model, and transcriptomic representations from a pretrained transcriptomic model with an additional trainable adapter. In doing so, this work demonstrably shows that this method, SemiClipped, allows morphological features to enrich the transcriptomic representations, and that this particular method outperforms other cross-modal distillation methods. Crucially, this allows this model to retrieve a greater number of known biological relationships from transcriptomic data, whilst also maintaining a high-level of interpretability at the level of genes.

In addition to proposing SemiClipped this work also proposes a novel augmentation method for transcriptomic data. Transformations which apply batch correction, in which samples are aligned to unperturbed control measurements, are applied to transcriptomic data to yield augmented samples, and simulate greater variance in the changes in experimental conditions under which transcriptomic data may be measured. This augmentation is shown to improve performance in biological retrieval tasks and maintains strong interpretability, a weakness of more common augmentation methods.

## Update after rebuttal
After reading all reviews and rebuttals, I have decided to leave my score unchanged. I welcome the additional figures and tables the authors have shared during rebuttal. To raise my score from a 4 to a 5, my assessment of the overall significance of the paper would have to be changed such that I felt this paper will have major impact on the community. This is more a question of the scope of the paper, and hence has not changed during rebuttal.

**Claims And Evidence:**

There are three key claims made in the article, namely:
1. that SemiClipped, the proposed method of cross-modal distillation, provides SOTA performance in the data limited regime
2.  that by freezing the encoder of the teacher modality (in this case morphological features) they prevent drift from student to teacher.
3. that they devise a novel biologically inspired data augmentation, PEA, that is capable of improving cross-modal distillation, and outperforms existing augmentations on benchmarks related to uncovering known biological relationships, and transcriptomic interpretability.

Each of these claims are well supported by evidence. Three datasets were used for OOD evaluations, reflecting generalisation to cell types, experimental conditions and gene expression quantification technologies. Each of these datasets represents a distribution-shift that is expected between training and inference in production, hence providing a realistic evaluation of performance. The claim that SemiClipped provides SOTA performance is demonstrated throughout the paper, as it is shown to perform better than a number of strong baseline models from the literature. By comparing, in Figure 6, the biological relationships that are recalled by a unimodal pretrained transcriptomic model with those recalled by a transcriptomic model trained via cross-modal distillation, the authors demonstrate clearly that i) cross-modal distillation can allow transcriptomic models to recall biological relationships typically found in microscopy representations ii) that current methods recall more relationships than unimodal models, but can lose some relationships captured before distillation and iii) that SemiClipped, which freezes the microscopy imaging encoder, recalls the most relationships overall and from those recalled by the unimodal transcriptomic model.

In combination with Figure 1, which shows that including a trainable adapter for both the imaging and transcriptomic models leads to poorer biological recall than using a frozen imaging module and trainable adapter for transcriptomics only, this supports the claim that SemiClipped provides SOTA performance and prevents drift from student to teacher.

Furthermore, Figure 2 demonstrates clearly that PEA i) provides consistent superior performance in the recall of biological relationships, and ii) in the transcriptomic interpretability of the representations for a number of benchmark augmentations in isolation and combination. The evaluation metrics for the biological recall task and transcriptomic interpretability are well explained in the Appendices. Including both of these evaluations provides a clear insight into how cross-modal distillation effects performance and interpretability.

**Essential References Not Discussed:**

I think it is worth mentioning [1] for demonstrating that encoders trained in SSL fashion with multi-modal data can outperform unimodal encoders in the limited data regime, with imaging and tabular data (which is somewhat related to transcriptomic data).


[1] Hager, P., Menten, M.J. and Rueckert, D., 2023. Best of both worlds: Multimodal contrastive learning with tabular and imaging data. In Proceedings of the IEEE/CVF Conference on Computer Vision and Pattern Recognition (pp. 23924-23935).

**Experimental Designs Or Analyses:**

I specifically checked the experiment for which results are shown in Figure 2, since these results support the key claims of this work. By using the three OOD evaluation datasets there were no concerns of data leakage between training and inference, and made comparisons between the proposed model and baseline models fair. I therefore see no issues with the design of the experiments used to form this figure and have confidence in these results.

**Methods And Evaluation Criteria:**

As mentioned above, the work focuses on two tasks, recall of known biological relationships and transcriptomic interpretability. The biological recall task is motivated by the problem at hand - in drug discovery a model that can predict changes in biological relationships from transcriptomic data is crucial for the automation of assessing the impact of the many compounds that exist in the space of all possible small molecules. This work focusses on cross-modal distillation with microscopy imaging, which provides rich insight into cell state, but lacks gene level interpretability. This motivates the transcriptomic interpretability task.

Additionally, by including three benchmark datasets that simulate real changes that one would expect between model training and deployment (changes in cell types, experimental conditions and gene expression quantification technologies) the author’s provide a realistic measure of model performance.

**Other Comments Or Suggestions:**

- Add y-labels to Figure 1, and Figure 2.
- Figure 3 took some time to parse, could this data be more clearly represented in a tabular format? I suppose the quantity of interest to highlight is the number of additional relationships gained via cross-modal distillation, and how many have been lost, and focussing on just these two may be enough to demonstrate the success of the method in a more immediate manner.

**Other Strengths And Weaknesses:**

This is a strong paper, with high quality authorship, clear and concise results, and well thought out experiments. The novel augmentation method for transcriptomics data could have impact alone, and it would be interesting to see how this could be adapted outside of a preclinical setting.

The main weakness of this work is the presentation of some of the results, there a few cases where figures are not well labelled. This is overcome by the clarity of the main text, but is a point that could be improved on.

**Questions For Authors:**

- Can you foresee the impact of PEA beyond transcriptomics data used in a preclinical setting, where datasets do not typically have a well defined notion of unperturbed controls?

**Relation To Broader Scientific Literature:**

It is well known that microscopy imaging can be combined with deep learning to extract useful features that describe cell phenotype, for example see [1]. However, while these models can be used to infer relationships between genes via comparing embeddings of cells perturbed by different gene KO, these models to not provide a direct transcriptomic interpretability, in the same way that a model utilising transcriptomic data would. By leveraging these modalities together, the growing power of microscopy models can be leveraged to create strong transcriptomic prediction models.


[1] Kraus, O., et al Masked autoencoders for microscopy are scalable learners of cellular biology. In CVPR, 2024.

**Theoretical Claims:**

This is not applicable for this work.

---

> ### Author Rebuttal · Authors · 2025-03-31
>
> We thank the reviewer T8LP for all their comments, and for their acknowledgement of the quality of the paper and the contribution. We aim to acknowledge and answer below the questions and suggestions of the reviewer :
>
> - __*“Additional reference” :*__ We thank the reviewer for pointing us toward this publication. As tabular data shares many common points with transcriptomics data, this merits an additional discussion on this paper in the final manuscript.
>
> - __*“there a few cases where figures are not well labelled” :*__ We agree with the reviewer, and while this was a limitation of the 8-pages format,, we will take advantage of the extra space in the final manuscript to ensure that all labels and captions are self contained and self explanatory.
>
> - __*“Add y-labels to Figure 1, and Figure 2.” :*__ We add y-labels to both figures in the final manuscript.
>
> - __*“Figure 3 took some time to parse, could this data be more clearly represented in a tabular format?” :*__ We thank the reviewer for their suggestion, this could clarify further the impact of our method. We have added to the final manuscript a new table that quantifies for each approach the amount of gain and loss of relationships through the distillation compared to other approaches. The added table is available in this anonymized link (https://imgur.com/a/qZB7q9l). We see that Semi-Clipped recalls the highest amount of known relationships, preserves the original transcriptomic information the best, and minimizes the loss of relationships the most.
>
> - __*“Can you foresee the impact of PEA beyond transcriptomics data used in a preclinical setting, where datasets do not typically have a well defined notion of unperturbed controls?” :*__ PEA’s augmentation principle is especially promising for biological data in batches with control, such as proteomics and metabolomics, where batch correction techniques are well established to mitigate technical variability. Although its impact may be more limited in contexts lacking a clear notion of unperturbed controls or where robust domain adaptation methods are already well established, PEA can still broaden the scope of existing solutions. PEA’s strategy of leveraging small and stochastic technical variations could extend and be adapted to specific post-processing methodologies  in areas such as signal processing (ECG, wearable devices) or astronomy. We will add a discussion on this question to the conclusion of the final manuscript.

---

### Official Review · Reviewer_rL4s · 2025-03-13

**Overall Recommendation:** 3

**Summary:**

The paper introduces Semi-Clipped, a method that transfers morphological information from microscopy images to transcriptomic data through cross-modal knowledge distillation. The authors adapt the CLIP loss by freezing a pretrained teacher encoder (for images) and learning a trainable adapter for transcriptomics. The authors also introduce PEA (Perturbation Embedding Augmentation), a data augmentation technique based on batch correction methods to introduce biologically plausible variation into the transcriptomic profile. Experiments on multiple out-of-distribution datasets (e.g., HUVEC-KO, LINCS, and SC-RPE1) are presented. The paper claims improved performance and enhanced biological signal relative to both unimodal baselines, existing cross-modal distillation approaches and augmentation techniques.

**Claims And Evidence:**

- The claim that Semi-Clipped achieves improved cross-modal knowledge distillation is convincingly demonstrated through comprehensive experiments and comparisons with multiple competitive baseline methods (including KD, SHAKE, VICReg, and others).
- PEA augmentation is interesting and generally improves the performance of multiple methods tested, the authors validated through statistically significant improvements.

**Essential References Not Discussed:**

N/A

**Experimental Designs Or Analyses:**

The choice of benchmark metrics is interesting. The experimental designs and analyses appear sound.

**Methods And Evaluation Criteria:**

- Although Figure 1 has motivated Semi-Clipped over vanilla clip, the paper could benefit from comparing the methods with Clip loss in distillation benchmark in Figure 2(a). The reviewer is concerned that Semi-Cipped outperforms Clip in a context dependent way, as the scGPT + Tx Adaptor + Image Adaptor is omitted in the Figure 1.
- There's a potential distribution shift issue since scVI was pretrained on single-cell data, while this study uses arrayed bulk sequencing data.

**Other Comments Or Suggestions:**

The paper adequately covers relevant literature but contains numerous citation errors,  the reviewer did not do an exhaustive check but the following are obviously wrong:

- Geneformer is incorrectly cited as "Chen, T. K., Wang, Z., Li, X., Li, Y., and Huang, K. Gene-former: A foundation model for generalizable gene ex-pression learning. bioRxiv, pp. 2023.01.14.524028, 2023.
  - instead of "Theodoris, Christina V., Ling Xiao, Anant Chopra, Mark D. Chaffin, Zeina R. Al Sayed, Matthew C. Hill, Helene Mantineo et al. "Transfer learning enables predictions in network biology." *Nature* 618, no. 7965 (2023): 616-624."
- scGPT is wrongly cited as: "Wang, Z., Song, B., Zhu, T., Li, B., Hu, Q., Tao, X., Chen, F., Wang, L., and Xie, P. scgpt: Transformer-based single-cell rna-seq data analysis. bioRxiv, pp.2023.02.24.529891, 2023."
  - which should be "Cui, Haotian, Chloe Wang, Hassaan Maan, Kuan Pang, Fengning Luo, Nan Duan, and Bo Wang. "scGPT: toward building a foundation model for single-cell multi-omics using generative AI." Nature Methods 21, no. 8 (2024): 1470-1480."
- Similar errors appear for scBERT, Drug-seq, and others

**Other Strengths And Weaknesses:**

N/A

**Questions For Authors:**

Can you please share your thoughts on previous sections?

**Relation To Broader Scientific Literature:**

The paper adequately discusses relevant works in the field.

**Theoretical Claims:**

N/A

---

> ### Author Rebuttal · Authors · 2025-03-31
>
> We thank the reviewer rL4s for their review, and for acknowledging the robustness demonstration of our claims. We will respond below to all their comments :
>
> - __*“The reviewer is concerned that Semi-Cipped outperforms Clip in a context dependent way” :*__ We thank the reviewer for pointing this out. To improve the demonstration of our claims, we add to Figure 1 of the final manuscript the performance of scGPT + Tx Adapter + Image Adapter. The revised figure can be found in this anonymized link (https://imgur.com/a/CDmKd3v). We show through this result that Semi-Clipped still outperforms Clip even in the scGPT context. To further illustrate this, we will add Clip as an additional approach in Figure 2.a in the final manuscript, to showcase with more clarity how Semi-Clipped compares to Vanilla Clip and other approaches.
>
> - __*“There's a potential distribution shift issue since scVI was pretrained on single-cell data, while this study uses arrayed bulk sequencing data” :*__ We thank the reviewer for pointing this out, and apologize for the confusion in the submitted manuscript. We do not have the distribution shift issue, as we pretrain scVI on arrayed bulk sequencing data (HUVEC-CMPD dataset), which is also used for the distillation training. While scVI was originally developed for single cell data, recent published benchmarks (e.g [Bendidi et al. 2024](https://arxiv.org/abs/2410.13956)) have shown that scVI still outperforms all models even when trained on Bulk sequencing data, which motivated our choice. We  have updated the final manuscript to better reflect and clarify this point, given the additional page of main content available.
>
> - __*“The paper adequately covers relevant literature but contains numerous citation errors” :*__ We sincerely thank the reviewer for pointing this out. We have now fixed all errors in all the paper’s citations, this will be reflected in the final manuscript.

---

### Official Review · Reviewer_rGLV · 2025-03-16

**Overall Recommendation:** 3

**Summary:**

This paper aims to extract representations of transcriptomics by distilling knowledge from microscopy images. The authors introduce (1) semi-clipped for cross-modal distillation from pretrained foundation models, and (2) perturbation embedding augmentation to generalize transcriptomics data.

**Claims And Evidence:**

The concept of 'semi-clipped' is not essentially different to the vanilla CLIP, but simply with fixed pre-trained microscopy image features using weakly pair samples.

**Essential References Not Discussed:**

[1] Spatially Resolved Gene Expression Prediction from Histology Images via Bi-modal Contrastive Learning. This paper may be relevant which uses CLIP for spatial transcriptomics learning.

**Experimental Designs Or Analyses:**

* No t-SNE to show the distribution of learnt features.
* Lack ablation study when teacher representations are not frozen.

**Methods And Evaluation Criteria:**

One significant concern: Microscopic images and transcriptomics data, although they may have shared information, but the feature spaces may not be highly overlapping. In other words, these two modals have complementary information, which cannot be mutually replaced. Forcing the transcriptomics data to extract only the features that are similar to the image data may cause the loss of distinct information of transcriptomics. This paper doesn't show empirically that the learnt representation of transcriptomics is relatively comprehensive.

**Other Comments Or Suggestions:**

N/A

**Other Strengths And Weaknesses:**

Well written and easy to follow, yet the methodology contribution is limited.

**Questions For Authors:**

N/A

**Relation To Broader Scientific Literature:**

N/A

**Theoretical Claims:**

No Theoretical contribution is made in this work.

---

> ### Author Rebuttal · Authors · 2025-03-31
>
> We thank reviewer rGLV for their review. We address reviewer’s comments below, in order to improve the clarity of the contribution of our submission :
>
> - __*"This paper doesn't show empirically that the learnt representation of transcriptomics is relatively comprehensive." :*__ We respectfully disagree. In the submitted manuscript, we proposed two different and complementary ways of evaluating empirically whether the learnt representation of transcriptomics through our approach is completely comprehensive : i) Both in Figure 2 and Table 1, we evaluate on the transcriptomics interpretability task, which is a published benchmark for evaluating whether the representation of transcriptomics is comprehensive and includes all information of the original transcriptomics data, through both reconstruction tasks and structural measures of data. ii) In figure 3, we focus on downstream tasks, and show that for discovering target relationships, our distilled transcriptomics representations uncover the same relationships the original transcriptomics data uncovers, while improving it further by uncovering imaging and other new relationships. Additionally, reviewer T8LP describes our evaluation as _“that SemiClipped, which freezes the microscopy imaging encoder, recalls the most relationships overall and from those recalled by the unimodal transcriptomic model.”_. This shows that we have evaluated the comprehensiveness of our transcriptomics representations thoroughly, and have shown that it conserves the original transcriptomics information even after distillation.
>
> - __*“t-SNE for representation distribution” :*__ We add in this anonymous link (https://imgur.com/a/GCi2FYj) a UMAP projection of our learnt features of the distillation model (bottom), compared to the transcriptomics features only (top). We compare both models at batch effect reduction (left, Mixed clusters is better) and perturbation separation (right, separated clusters is better). While both approaches are good at reducing batch effect, the distillation approach is largely better at separating different perturbations. This is also reflected in the NMI metric for clustering the different perturbations with a K-Means (0.128 for the transcriptomics only, 0.481 for the distillation). We had previously hesitated to add UMAP/t-SNE views due to their anecdotal nature. We will however add this figure to the final manuscript for enhanced clarity.
>
> - __*"Lack of ablation study when teacher representations are not frozen." :*__ We respectfully disagree.  In the submitted manuscript, we show in Figure 1 that unfreezing the teacher representations through adding an Image Adapter leads to a drop in performance, likely due to modality drift. We add an additional evaluation of scGPT used with an Image Adapter in our response to reviewer rL4s for more comprehensiveness.
>
> - __*“Proposed reference” :*__ We thank the reviewer for the reference, we will include it in the final manuscript in the related works section (Biologically Relevant Representations).
>
> - __*“The methodology contribution is limited” :*__ We wish to point out that in addition to being the first to prove that distillation from Imaging to Transcriptomics is possible, we have proposed PEA, a completely novel and new approach for data augmentation on omics data that preserves biological information for multimodal learning, tackling one of the main problems of the omics field : Limited paired data and lack of biology preserving data augmentations. We also want to point out that reviewer rL4s described our claim and approach as *“PEA augmentation is interesting and generally improves the performance”*, while reviewer T8LP describes our approach as *“This is a strong paper, with high quality authorship, clear and concise results, and well thought out experiments. The novel augmentation method for transcriptomics data could have impact alone.“* and *“By leveraging these modalities together, the growing power of microscopy models can be leveraged to create strong transcriptomic prediction models.”*. Reviewer DZgQ described our work as *“For the first time, batch correction is restructured as data augmentation, addressing the scarcity of biological multimodal data while ensuring performance improvement with interpretability.”*. This being an application track submission for ICML, where submissions are judged by their real world relevance and the robustness of the claims and experiments, we hope this clarifies the reviewer’s assumptions about our work.

---

### Decision · Program_Chairs · 2025-05-01

**Decision:**

Accept (poster)

**Comment:**

This paper discovers that transcriptomics and microscopy images are complementary modalities, and proposes a methodology to learn representations of transcriptomics by distilling knowledge from microscopy images. The reviewers had concerns regarding limited novelty of semi-clipped, missing ablation and visualization, no biological grounding for PEA and computational cost. The authors addressed most of the concerns, and the strengths of the paper outweighs the drawbacks.